# Assessing Excoriation (Skin-Picking) Disorder: Clinical Recommendations and Preliminary Examination of a Comprehensive Interview

**DOI:** 10.3390/ijerph19116717

**Published:** 2022-05-31

**Authors:** Ivar Snorrason, Han-Joo Lee

**Affiliations:** 1Center for OCD & Related Disorders (CORD), Massachusetts General Hospital, 185 Cambridge Street, Boston, MA 02114, USA; 2Department of Psychiatry, Harvard Medical School, Boston, MA 02115, USA; 3Department of Psychology, University of Wisconsin-Milwaukee, 2441 E Hartford Ave, Milwaukee, WI 53211, USA; leehj@uwm.edu

**Keywords:** skin picking, subtypes, excoriation, interview, psychometric, students, DSM-5

## Abstract

Excoriation (skin-picking) disorder (SPD) is a psychiatric condition with variable clinical presentation. We developed the Diagnostic Interview for Skin Picking Problems (DISP), a semi-structured interview designed to assess (1) DSM-5 criteria for SPD and (2) several clinical features of SPD, including the frequency and duration of picking episodes, and the proportion of time devoted to picking at different body areas. The DISP was administered along with other measures to 120 college students (88% women, average age = 22 years) with suspected skin picking problems (based on their responses to a screening survey). The results showed that the DISP had good inter-rater reliability, test–retest reliability over 1–5 months, and convergent/divergent validity. We also found that participants displayed divergent clinical characteristics depending on the pattern of frequency, duration, and body location of picking episodes (e.g., those who primarily picked at the fingers had a unique clinical presentation). Overall, the findings provide preliminary support for the psychometric properties and clinical utility of DISP. The results also underscore the importance of accurately assessing frequency, duration, and body location of picking episodes.

## 1. Introduction

Excoriation (skin-picking) disorder (SPD) is a potentially severe psychiatric condition that is classified as an obsessive compulsive and related disorder in the fifth edition of the diagnostic and statistical manual of mental disorders (DSM-5; [1]). SPD is often thought to be closely related to hair pulling disorder (trichotillomania) [2] and other body focused repetitive behaviors such as excessive nail biting and cheek biting [3]. The DSM-5 criteria [1] define SPD as a recurrent skin picking that causes skin lesions and is associated with clinically significant distress or impairment in functioning, as well as failed attempts at stopping or decreasing the behavior. Exclusionary criteria rule out picking that is solely due to a medical condition (e.g., dermatological problems), another psychiatric disorder (e.g., body dysmorphic disorder) or the effects of substances (e.g., stimulants).

SPD has received limited empirical attention, but emerging literature suggests the problem can become quite severe [4] and is more prevalent than previously thought [5]. In a national survey of U.S. adults [6], 16% of responders endorsed recurrent picking that caused skin damage, and 1.4% met full DSM-5 criteria for SPD. Surveys among college students have shown that about 2% of participants meet criteria for SPD [5,7,8]. In treatment-seeking samples, the majority of those with SPD are females (75–94%), and the problem tends to begin in adolescence and have a chronic course if untreated [2].

Although semi-structured interviews are often utilized in research to establish DSM diagnosis of SPD, published descriptions and psychometric analyses of such instruments are limited [9,10]. Additionally, DSM criteria for psychiatric disorders typically represent a “bare bones” definition of the condition [11,12]. As such, they may have suboptimal content validity—i.e., they do not fully capture all relevant aspects of the target construct [13]. It can therefore be useful to develop an instrument that captures DSM-5 definition of SPD, as well as other clinical features. We suggest that in addition to the DSM-5 criteria, two clinical features are particularly useful in the assessment of SPD.

First, it is important to accurately assess how much time the person spends picking at their skin. According to the DSM-5 criteria, “recurrent” skin picking is required for diagnosis of SPD, but the exact amount is not specified. Nevertheless, accurate assessment of time devoted to picking can be informative, as this can help estimate the severity of the problem. One strategy is to assess the overall duration or amount of picking (e.g., the number of minutes/hours spent picking on an average day). However, some individuals report frequent and short episodes (e.g., several 1 min episodes throughout the day), whereas others report less frequent episodes that last for hours (e.g., one 2 h episode every evening) [7,14]. Thus, simply assessing the overall duration of picking may not give an accurate picture of the problem. We therefore recommend assessing not only the overall duration of skin picking but also the frequency and duration of episodes.

Secondly, it is important to assess what area of the body the person picks at. Skin picking can occur anywhere on the body, and patients often pick multiple body areas [15,16]. However, individuals with SPD frequently report a strong preference for a particular site [15,16]. Thus, simply asking patients to list all their picking targets will not give a complete picture of the problem. We recommend documenting all body area as well as the proportion of time devoted to each area.

One reason it is helpful to assess time devoted to each body target is that clinical features of SPD may differ depending on the primary picking targets. For example, individuals who pick at the face are often more concerned with the effect picking has on appearance of the skin than individuals who pick at other parts of the body [17]. Additionally, studies suggest that different picking sites may predict different comorbidity patterns [18]. For example, fingerpicking is common among those with SPD [15,16] and evidence suggest that those who pick at their fingers are more likely to have problems with nail biting compared with those who do not pick their fingers [18]. Additionally, people who pick at the fingers may engage in greater number of brief picking episodes, but experience less psychosocial impairment due to picking, compared to people who pick other areas [18]. However, this issue has received limited research attention. If fingerpicking is a common subtype of SPD that has unique clinical characteristics (e.g., different response to treatment), it is important that the research literature documents the extent to which study samples are made up of individuals who primarily pick at their fingers (and other picking sites). Unfortunately, studies rarely report proportion of the sample who primarily pick at different body areas. Moreover, we are not aware of any study that reported the proportion of time devoted to picking at each body area.

In this study, we developed the Diagnostic Interview for Skin Picking Problems (DISP) and examined its psychometric properties among college students with SPD. Validating a diagnostic interview in a college student sample is important because large proportion of research in SPD occur in student populations [4,5,7,8]. The DISP is a semi-structured interview designed to assess (1) DSM-5 diagnosis of SPD and (2) clinical features of SPD, including specific symptom presentation (e.g., picking sites, and frequency of picking episodes), phenomenology (e.g., picking urges and awareness of picking) and the course of the problem (e.g., age at onset, abstinence periods). We administered the DISP to a sample of college students with suspected SPD (based on their responses to an online self-report screening survey) and examined inter-rater reliability, test–retest reliability, and construct validity.

As noted above, different bodily target may be associated with unique clinical features. We therefore divided the sample in three subgroups depending on primary bodily target: (1) fingers, (2) face and (3) other body areas. We then compared clinical features across the three groups. We expected that individuals with primary finger picking would report more frequent and brief picking episodes, but less distress and impairment, compared to the other two groups.

## 2. Materials and Methods

### 2.1. Participants

Participants were 120 undergraduate students at a university in the midwestern U.S. who had endorsed problematic skin picking in an online screening survey and subsequently participated in an in-person assessment. The sample included 105 females (87.5%) and 15 males (12.5%). The average age was 22.1 year (SD = 4.1; range 18 to 37 years). The majority (84.2%) identified as White, 3.3% as Asian, 1.7% as Black/African American, and 2.5% as multi-racial. The majority (92.7%) reported non-Hispanic ethnic origin.

### 2.2. Measures

*Online Pre-Screener for SPD*. We used a multiple-choice question as an online pre-screener to identify potential participants: “Have you ever had the habit of excessively picking at your skin?” Response options were: (a) “No never”, (b) “No (I sometimes pick skin, but never excessively)”, (c) “Yes, in the past I used to pick skin excessively (no picking the past month or longer)”, and (d) “Yes, currently, I excessively pick skin, and it causes problems”. Responders who endorsed option d were invited to participate in an in-person assessment.

*The Diagnostic Interview for Skin Picking Problems (DISP)*. The DISP is a 16-item semi-structured diagnostic interview designed to assess diagnostic criteria and clinical features of SPD. The items are organized in four domains: (1) DSM-5 diagnosis, (2) symptom presentation, (3) phenomenology, and (4) course of the disorder. The items were developed based on review of the literature [2] and clinical experience of the authors. The DISP can be download here: https://osf.io/t6ycb/.

Screening Items. Item #1 is a dichotomous (Yes/No) screening item that establishes history of excessive picking habit. Individuals who endorse the screening items are asked all the other questions. Item #2 establishes if the individual currently has the habit (past 30 days) or had it in the past. These items are designed to enable the efficient administration of the DISP in case of being used as a stand-alone instrument without a preceding screening procedure.

Frequency/duration of episodes and picking sites. Item #3 documents where on the body the skin is picked. The interviewee is asked to list all the body areas he or she picks at and then estimate the proportion of time that is devoted to each site (“In percentages, how much of your picking time is devoted to …, and how much to …”, etc.). For example, an individual may pick the face and the fingers, but 80% of the total picking time is devoted to the fingers, and 20% to the face.

Item #4 asks the interviewee to estimate the frequency and duration of picking occasions (“How often do you pick your skin?”, “On average, how many episodes per day/week/month and how long does each episode last?”). To index the overall duration of skin picking, the number of episodes per day was multiplied by the average number of minutes per episode (DISP Total Duration). (In the rare cases where the individual picked weekly, the number of episodes reported are divided by 7; thus, it reflects episodes per day.)

DSM-5 Diagnostic items. Items 5 to 7 are dichotomous (Yes/No) diagnostic items that help determine a DSM-5 diagnosis. Specifically, these items assess whether the skin picking habit causes skin damage (item #5), subjective distress (item #6), interference with functioning (item #7), or repeated quit attempts (item #8). Items #9–11 help rule out other causes of skin picking. Item #9 rules out skin picking that is solely due to the use of substances (e.g., cocaine). Item#10 helps rule out skin picking habit that is solely cause by a medical or dermatological condition. Some interviewees may report picking at skin imperfections caused by dermatological conditions such acne, eczema, folliculitis (inflamed hair follicles), or keratosis pilaris (small bumps typically on the upper arm or thighs). In these cases, it is important to determine if the picking habit is solely caused by the dermatological condition (e.g., “Do you think you would not pick at all if you did not have the dermatological condition?”, “Did you ever stop picking after the dermatological condition subsided?”, “Do you pick at body areas that are not affected by the condition?”).

Item #11 aims to exclude skin picking that is solely due to appearance concerns in body dysmorphic disorder (BDD). The interviewer should make sure that the appearance concerns are exaggerated in case they meet the diagnostic criteria for BDD (i.e., “not observable or appear slight to others”, [1] (p. 242)). In other words, it needs to be determined that these are not simply reasonable concerns about skin damage caused by picking. Additionally, the interviewer should determine if the appearance concerns are the main motivation for picking. Many individuals with SPD report desire to smooth out skin imperfections and feel they are “fixing” the skin when picking. This is not the same as picking in response to excessive concerns about having abnormal, ugly or unattractive skin. Only the latter excludes a diagnosis of SPD [1,18,19].

Phenomenology and Course of the Disorder. Items #12 and #13 reflect the DSM-IV criteria for impulse control disorder (Do you get an urge or an impulse to pick? Do you experience relief or gratification when you pick?). Item #14 assesses how often the individual is unaware of the behavior as it occurs (Do you ever pick without noticing it? “In percentages how often are you NOT aware of it?”). Items #15 and #16 ask about age at onset and chronicity, respectively.

*Convergent Measure*. In order to investigate convergent validity of the DISP, participants were asked to complete a previously validated self-report measure of SPD—the Skin Picking Scale-Revised (SPS-R; [15]). The SPS-R is an eight item self-report scale that measures severity of SPD over the past week on two subscales. The Symptom Severity scale includes items that assess (1) duration of picking urges per day, (2) intensity of picking urges, (3) duration of picking behavior per day, and (4) the degree of control over the behavior. The Impairment Scale assesses negative consequences of skin picking in four domains: (5) emotional distress, (6) functional impairment, (7) social avoidance and (8) skin damage. All items are rated on 5-point scales ranging from 0 (e.g., none) to 4 (e.g., extreme). Previous studies have documented good psychometric qualities of the SPS-R, including high internal consistency, robust two-factor structure, and adequate convergent/discriminant validity [20,21]. In the current sample, the internal consistency of SPS-R was good (total score α = 0.84; Severity subscale α = 0.86; Impairment subscale α = 0.79). We examined the correspondence between DISP DSM-5 diagnosis and the SPS-R Impairment scale, and between the DISP frequency/duration items/indices and item#3 on the SPS-R (average duration per day).

*Divergent Measures*. To examine the divergent validity of DISP, participants completed three measures of related but distinguishable constructs. First, although skin picking causes skin damage, DISP should not have strong correlation with a measure of general dermatological conditions. Thus, we administered the Current Skin Complaints Scale (CSCS; [22]), which is a brief self-report measure designed to assess the presence of common skin complaints. Participants are asked if they have had (1) pimples, (2) dry skin, (3) rash, and (4) itchy skin during the past week. Response options are “No”, “Yes, a little”, “Yes, quite a lot”, and “Yes, very much”. In the current study, we examine the correlation between the sum of the four CSCS items and the DISP items/indices. The internal consistency of the CSCS items in the current sample was modest (α = 0.63).

Second, although research has shown association between negative affect and skin picking, the DISP should be relatively unrelated to a measure of general negative affect. To examine negative affect, we asked participants to complete the Depression Anxiety Stress Scale 21-item version (DASS-21; [23]). The DASS-21 is a widely used 21-item questionnaire designed to measure symptoms of depression, anxiety, and stress over the past month. Each domain contains seven items in which respondents are asked to indicate the extent to which a statement applies to them using a 4-point Likert scale (0–3). Studies have shown that the three scales have good psychometric properties but also appear to tap a shared general dimension of negative affect [24]. We examined the relationship between DISP and the DASS-21 scales all of which had good internal reliability in the current sample (Stress α = 0.87; Anxiety α = 0.82; Depression α = 0.92).

Third, we examined the association between obsessive compulsive disorder symptoms and DISP. Obsessive compulsive disorder (OCD) and SPD had distinct clinical features and there should be limited correlation between DISP and measures of OC symptoms. To assess OC symptoms, we administered the Obsessive Compulsive Inventory-Revised (OCI-R; [25]), which is a validated 18 item self-report scale that measures the severity of OC symptoms over the past month. The scale has six subscales reflecting different OC symptoms: checking, washing, obsessing, neutralizing, hoarding, and ordering. Each item is rated on a 4-point scale from 0 to 3. The OCI-R has been shown to have good psychometric properties in clinical and non-clinical samples [25]. The OCI-R had excellent internal consistency in the current sample (α = 0.92).

### 2.3. Procedure

*Data collection*. Participants were recruited from a large pool of students from the university. Each semester all students attending classes in the psychology department are offered to complete a screening survey that determines eligibility for various research studies conducted in the department (via the SONA research management system). Over six semesters, a screening item for problematic skin picking was included in the survey. Responders who endorsed current skin picking problems were sent invitations (by email) to participate in an in-person assessment for research studies on SPD. All interviews were conducted in a quiet room located in a psychology laboratory at the university. During the in-person assessment, participants underwent the DISP and completed the self-report measures. Informed consent was obtained at the outset of the session and the students received extra credit for participating. This study was approved by the Institutional Review Board of the University.

Given the subjective nature of some of the DISP items (e.g., functional impairment due to picking), it is important to demonstrate that they can be scored consistently across interviewers. We therefore obtained data to conduct inter-rater reliability analysis in a subsample of interviews. This was achieved by having a second interviewer sit in the interview room and rate the DISP independently along with the primary interviewer. Additionally, to show that DISP yields consistent results across administrations, we obtained data for test–retest reliability analysis by contacting a subsample of participants 1 to 5 months after the study session and administered the DISP again via telephone.

*Interviewers*. The first author conducted 87.5% (*n* = 105) of the in-person interviews and all the follow-up phone interviews. An undergraduate research assistant, who had received training in administering the DISP, conducted 12.5% (*n* = 15) of the interviews. (The training included shadowing the first author during interviews and conducting interviews under live supervision.) The inter-rater reliability was examined in a subsample of interviews that were conducted by the first author and independently rated by an experienced interviewer who had received training in conducting the DISP. (Data from the training interviews were not included in the inter-rater analyses.)

### 2.4. Data Analyses

The hypotheses and the analytic plan were not preregistered. We examined descriptive statistics (e.g., frequencies, means and standard deviations) for clinical variables among participants who met DSM-5 criteria for SPD. In order to examine inter-rater and test–retest reliability of dichotomous items, we calculated kappa (k) statistic. When marginal distribution was very uneven (>70% in one cell) we used prevalence-adjusted kappa [26]. To examine inter-rater and test–retest reliability for continuous items, we calculated two-way mixed intra class coefficient. For construct validity analyses, we examined how strongly the DISP items/indices correlated with a measure of the same construct (SPS-R) and measures of overlapping but distinct constructs (i.e., DASS-21, CSC, and OCI-R). We calculated Pearson correlation coefficient between two continuous variables, and biserial correlation coefficient between dichotomous and continuous variables. We used an online calculator (https://www.psychometrica.de/correlation.html (accessed on 30 May 2022)) to determine if size of two correlations were significantly different from each other (see [27]). We explored the differences between three subgroups of participants with different primary picking sites using one-way analyses of variance (ANOVA). When significant differences emerged, Tukey’s post hoc test procedure was used to determine what groups differed.

## 3. Results

### 3.1. Clinical Characteristics

Below we report descriptive statistics for the clinical characteristics items among those participants who met the full DSM-5 criteria for SPD in the interview (*n* = 73).

#### 3.1.1. Picking Sites

About two-thirds (*n* = 49; 67.1%) of the sample reported picking more than one body area (M = 2.3; SD = 1.2, range 1 to 5). Table 1 shows the number of participants that reported different body areas.

#### 3.1.2. Picking Frequency/Duration

All participants picked skin every day (*n* = 61; 87%) or every other day (i.e., at least four times per week; *n* = 9; 13%). The average total duration of picking was 55.8 min per day (SD = 66.3), with an average of 6.5 episodes per day (SD = 8.3) and an average of 16.9 min per episode (SD = 23.3).

#### 3.1.3. Phenomenology

Most of the sample reported experiencing an urge or an impulse to pick (79%), or gratification/relief during the act of picking (89%). About a third (31%) reported picking without awareness more than 50% of the time, another third about 50% of the time, and a third less than 50% of the time. The average was 42% of the time (SD = 32%).

#### 3.1.4. Age at Onset and Course of the Problem

The average age of onset was 11.2 years (SD = 5.6, range 2 to 24 years) and the average duration of the problem was 11.5 years (SD = 6.5, range 1 to 33 years). About half (54%) had never stopped picking for more than few days, and the majority (93%) had never stopped picking for more than few months. Both the median and the mode of the distribution for the longest abstinence period was seven days (M = 53; SD = 138.3; ranged from few hours to 2 years).

### 3.2. Diagnostic Items

Table 2 shows the proportion of the sample that endorsed each of the diagnostic items. Out of the 120 individuals who participated in an in-person assessment, eight (6.6%) did not report current excessive skin picking in the DISP (despite endorsing current skin picking problems in the screening survey). Thereof, two reported past skin picking problems (no excessive picking the past 30 days or more), three mostly picked at the nails, one had a skin biting/gnawing habit, and two reported negligible skin picking behaviors.

Of those 112 who did report current excessive skin picking, 92% typically suffered skin damage, 94% wanted to stop or decrease skin picking, 77% had attempted to stop, 79% experienced emotional distress due to the habit, and 44% impairment in functioning. Four individuals (3.6%) were excluded because their picking habit was judged to be solely due to a dermatological problem (e.g., eczema). Another six individuals reported picking at skin imperfections caused by a dermatological condition (e.g., acne). However, they were not excluded because the condition was not the sole reason for their picking behaviors (e.g., picked at other body areas as well). Two participants were excluded because their skin picking was better accounted for by a diagnosis of BDD. No participant reported solely picking due to the use of substances, although one had developed a picking habit when addicted to methamphetamine but had continued picking after becoming sober three years earlier. (Another participant reported worsening of symptoms when taking Adderall, but also picked excessively when not taking it.) A total of 106 participants (88%) reported current excessive skin picking that was not due to a medical condition, another psychiatric problem or drug use, and 73 (60.8%) met full DSM-5 criteria for SPD. Table 3 shows correlations between each diagnostic item and full DSM diagnostic status. Prior quit attempts and emotional distress due to skin picking had the largest correlation with the diagnostic status in the sample.

### 3.3. Inter-Rater and Test–Retest Reliability

We examined inter-rater reliability for all the DISP items between two raters in a subset of interviews (*n* = 17). The agreement ranged from excellent to perfect (Table 3). In another subsample (*n* = 21) we examined test–retest reliability of all the DISP items over one to five months (mean 92.2 days; SD = 43.2; range 35 to 156 days). The test–retest reliability ranged from adequate to perfect (Table 3).

### 3.4. Construct Validity

The divergent and convergent validity was explored in a subsample of participant who endorsed current excessive skin picking not better accounted for by dermatological problems, drug/medication use or BDD (*n* = 106). We examined to what extent the DISP items/indices correlate with a previously validated measure of the SPD (SPS-R) and measures of overlapping but distinct constructs (negative affect, obsessive compulsive symptoms and general skin complaints). As shown in Table 4, the DISP items/indices did not have significant correlations with the divergent measure, except DSM-5 diagnosis had small correlation with stress (r = 0.21) and depression (r = 0.20). Table 4 also shows that the DISP items/indices were correlated with convergent measure (SPS-R) in predictable ways. Specificity, there was a moderately large positive correlation (r = 0.42) between DSM-5 diagnosis and SPS-impairment. A correlation of 0.42 is high considering that the two instruments cover different periods (past week vs. the past month), and this correlation was significantly larger than the correlation between DSM-5 diagnosis and stress (r = 0.42 vs. r = 0.21, z = 1.9, *p* < 0.05) or depression (r = 0.42 vs. r = 0.21; z = 1.8, *p* < 0.05). We also found a moderate positive correlation between the DISP frequency/duration items/indices and item 3 on the SPS-R, which measures average time spent picking per day over the past week. As would be expected, the DISP Total Duration had the highest correlation with this item (r = 0.41).

### 3.5. Correlates of Primary Picking Sites

Previous research [17,18] indicates that picking of different body areas can be associated with different clinical features. We therefore examined clinical correlates of picking sites among the 106 individuals who endorsed current excessive skin picking that was not caused by dermatological problems, drug/medication use or BDD. We created three groups based on the most common picking sites: (1) fingers, (2) face and (3) body (i.e., all sites except face or fingers). The three groups were discrete, and each participant were assigned to a group based on the picking site they devoted the most time to.

Table 5 shows the DISP variables across the three groups. The groups were relatively homogeneous with respect to the primary picking site, which supports the classification method. Specifically, 87% of the picking duration in the fingerpicking group was devoted to the fingers, 79% of the picking duration in the face picking group was devoted to the face, and 82% of the picking duration in the body picking group was devoted to the body. Group comparisons showed that those in the fingerpicking group differed from the other two groups on several clinical features. First, the face and body groups reported significantly greater functional impairment and emotional distress compared with the fingerpicking group (although the difference for emotional distress was only marginally significant from those with primarily body picking, *p* = 0.06). Secondly, the face and body groups reported greater number of picking areas relative to the fingerpicking group. Thirdly, compared to the fingerpicking group, the other two groups reported fewer but longer picking episodes and greater awareness of the behavior.

## 4. Discussion

In this study, we examined the psychometric properties of the DISP, a semi-structured diagnostic interview that assesses clinical features and DSM-5 diagnosis of SPD. In addition to helping establish DSM-5 diagnosis of SPD, the interview helps the interviewer collect important information on clinical features of SPD, including detailed information on frequency, duration, and body location of picking episodes. Overall, the results support the psychometric quality of the interview and underscore the usefulness of comprehensively assessing clinical features of SPD beyond the DSM-5 criteria, including frequency, duration, and body location of picking

### 4.1. Psychometric Properties

The results provided preliminary support for the psychometric properties of the interview. We found evidence for convergent validity as core items/indices of this new instrument (e.g., DSM-5 diagnosis and total picking duration) had moderate correspondence with a previously validated measure of the same constructs (the SPS-R). We also found evidence for discriminant validity as these core DISP items/indices were relatively unrelated to valid measures of overlapping but distinct constructs, including negative affect, obsessive compulsive symptoms, and skin complaints. Additionally, the results showed excellent inter-rater agreement for each of the diagnostic item as well as the full DSM-5 SPD diagnosis. The high agreement was likely aided by to the use of unambiguous questions that included several disorder-relevant examples. We believe the DISP can be administered in a reliable fashion by bachelor level interviewers who have received appropriate training. We also examined the test–retest reliability of DISP over 1 to 5 months. SPD is usually a chronic condition, and it can be expected that diagnostic features remain relatively stable over this period. The test–retest reliability ranged from satisfactory to perfect, which suggests the instrument will yield data that are consistent across multiple administrations.

### 4.2. Assessment of Clinical Features beyond the DSM-5 Criteria

Our data underscore the importance of assessing the number and length of picking episodes (not just total duration) because different patterns can be associated with unique clinical features. For example, in our sample, the length of episodes, but not frequency of episodes, was associated with negative affect (although the correlation was not significant in all cases). The results also supported the prediction that different picking sites can moderate clinical features [18,19] and it is therefore important to not only document number of body areas targeted, but also time devoted to each area. Specifically, we found that individuals who primarily picked at the face or body were more likely than individuals who picked primarily picked at the fingers to report distress and impairment due to SPD and tended to pick at more body areas. These participants were also more likely to report longer picking episodes and had more reflective awareness of the picking behavior compared to those who picked primarily at the fingers. These findings are consistent with clinical impression that suggest individuals who primarily pick at their fingers often engage in high frequency low intensity picking (presumably because they can pick in most situations).

### 4.3. External Validity

Using a screening survey among college students was effective in identifying individuals with SPD. Of those who endorsed the online screening item and presented to the testing session, 88% (106/120) were judged to have an excessive skin picking habit not better explained by drug use, BDD or dermatological problems, and 61% (73/120) met DSM-5 diagnostic criteria for SPD. Overall, the results suggest that DISP can provide a reliable and valid assessment of SPD diagnosis and symptoms in college student populations. However, further studies are needed to examine the extent to which the findings (e.g., clinical features of finger picking) generalize to treatment seeking samples.

In order to gauge the external clinical validity of the results, we examined the clinical characteristics of those participants who met full DSM-5 criteria for SPD. In our sample, the average total picking duration per day (57 min) was slightly less than the average reported in previous clinical samples (ranging from 84 to 109 min; [14,15,16]). However, we assessed duration by multiplying the number of episodes per day with minutes per episode. This method may produce a more conservative estimate than the usual method of having participant directly report the total duration. On the other hand, it also seems likely that the current sample on average presented with less severe problems than treatment-seeking clinical samples.

Nonetheless, all other clinical characteristics in our sample were highly similar to that found in treatment seeking populations (for review see [2]). The majority of participants in our sample reported a daily picking habit that had persisted for years without long periods of abstinence. This is consistent with the typical chronic and persistent course of SPD documented in clinical samples [2]. Additionally, similar to our data, most studies in psychiatric samples have reported average age at onset in early adolescence, and around 1:9 male to female ratio [2]. Furthermore, our results were similar to clinical samples with respect to the body areas picked at, the degree of awareness reported and the proportions of individuals who endorsed urge/impulse prior to picking and gratification/relief during picking [16]. Overall, the similarities between our sample of students meeting DSM-5 criteria and clinical SPD samples provide justification for recruiting SPD subjects in university settings. However, we also found that many participants did not meet full DSM-5 criteria, which underscores the importance of using diagnostic interviews to verify SPD diagnosis.

### 4.4. Limitations, Future Research and Suggested Refinements

Our study had some limitations and lead to several suggestions for future research and refinement of the instrument. First, despite the relevance and preponderance of student sample-based research for SPD, it is important to examine the suitability of the DISP for treatment seeking populations and document its psychometric qualities in clinical settings. A limitation of the DISP is a lack of exclusionary item for excessive skin picking related to psychotic symptoms (e.g., delusional parasitosis). Although this is a rare phenomenon [14] it may be advisable to consider adding an exclusionary item to the DISP, especially when it is administered to clinical populations (e.g., “Do you mainly pick because you feel there is something foreign in or under your skin, e.g., insects, bugs, or fibers?”). Secondly, our data underscored the importance of assessing not only picking sites but also the proportion of time the individual devotes to each site out of the total time spent picking. We assessed this by having participants list their picking sites and estimate the percentage of time devoted to each. From these data, we could extrapolate the time spent picking each body area (i.e., by converting the percentage into a fraction and multiply it with overall picking duration). Another approach would be to ask participants to directly estimate the length and number of episodes for each picking site. Although more time consuming, this method could give a more accurate clinical picture and allow a more direct assessment of time devoted to each area. Third, the ratings of urge prior to picking and gratification/relief during picking may be better assessed on a dimensional scale. One option would be to assess these symptoms in the same fashion as the awareness of picking (e.g., “in percentages, how often do you experience urge prior to picking skin?”). Fourth, even though DISP is a relatively comprehensive interview, it does not assess potentially important variables that contribute to the cause or the maintenance of skin picking problems (e.g., emotional or contextual factors). Additionally, future investigations should examine whether the interview can be utilized to sensitively detect change in treatment-related outcomes. Fifth, our data and other research indicate that SPD often occurs in childhood, and future research is needed to adapt the DISP to pediatric populations. Finally, the sample size was relatively small, particularly for the analysis of inter-rater and test–retest reliability. Thus, the study needs to be replicated in larger samples. Similarly, the sample consisted primarily of non-Hispanic White college students, and the study should be replicated in more socioeconomically and ethnically diverse samples. Additionally, the representation of male participants was quite small, and further studies are needed to better understand whether DISP will demonstrate good psychometric properties and clinical utilities across genders.

## 5. Conclusions

The DISP is a comprehensive diagnostic interview designed to assess DSM-5 diagnosis and clinical features of SPD. This initial study suggests that the DISP has good inter-rater and test–retest reliability as well as acceptable construct validity among college students with SPD. We believe the DISP can be used in general and clinical settings by trained interviewers (our data suggest that bachelor level interviewers can reliably administer the DISP). The results highlight the utility of going beyond the DSM-5 criteria and accurately assess the duration, frequency, and body location of picking episodes. This type of assessment allows researchers and clinicians to identify clinically meaningful subgroups of individuals with SPD. Specifically, the results shows that individuals who primarily pick at the fingers show unique clinical features, including a relatively less severe clinical problem. Some refinement of individual items may further improve the instrument and more studies are needed to examine psychometric features in treatment seeking populations.

## Figures and Tables

**Table 1 ijerph-19-06717-t001:** Picking sites among participant with SPD (*n* = 73).

Body Areas	*n* (%)
Face	45 (61.6)
Fingers	31 (42.5)
Legs	18 (24.7)
Back	16 (21.9)
Arms	16 (21.9)
Chest	9 (12.3)
Shoulders	8 (11.0)
Scalp/head	8 (11.0)
Lips	8 (11.0)
Feet/Toes	4 (5.5)
Hands	2 (2.7)
Thighs	1 (1.4)
Neck	1 (1.4)

**Table 2 ijerph-19-06717-t002:** Proportion of the sample that endorsed each of the diagnostic items and their correlation with full DSM-5 criteria.

Item Content	Proportion (%) Endorsed	Correlation with DSM-5 Diagnosis ^1^
Excessive skin picking	112/120 (93.3%)	0.33 **
Skin damage	103/112 (92.0%)	0.42 **
Desire to stop picking ^2^	78/83 (94%)	0.27 *
Attempts at stop picking	86/112 (76.8%)	0.65 **
Distress due to picking	88/112 (78.6%)	0.63 **
Impairment due to picking	49/112 (43.8%)	0.41 **
Picking due to medical problem	4/112 (3.6%)	---
Picking solely due to body dysmorphic disorder	2/112 (1.8%)	---
Picking solely due to the use of substances	0/112 (0.0%)	---

^1^ Biserial Correlation Coefficient; ^2^ This item was not included at the beginning of the data collection and therefore only administered to 83 individuals; * *p* < 0.05; ** *p* < 0.001.

**Table 3 ijerph-19-06717-t003:** Inter-rater and test–retest reliability.

	Inter-Rater Reliability (*n* = 17)	Test–Retest Reliability (*n* = 21)
** *Diagnostic Items* **		
Excessive skin picking	1.00 ^b^ *	1.00 ^b^ *
Skin damage	0.77 ^b^ *	1.00 ^b^ *
Desire to stop	1.00 ^b^ *	0.85 ^b^ *
Attempts to stop	1.00 ^b^ *	0.90 ^b^ *
Emotional distress	1.00 ^b^ *	0.71 ^b^ *
Functional impairment	0.88 ^a^ *	0.89 ^a^ *
Dermatological problem	1.00 ^b^ *	1.00 ^b^ *
Body dysmorphic disorder	1.00 ^b^ *	1.00 ^b^ *
Substance use	1.00 ^b^ *	1.00 ^b^ *
Full SPD DSM-5 diagnosis	1.00 ^b^ *	0.74 ^a^ *
** *Symptom Presentation Items* **		
N of picking sites	1.00 ^c^ *	0.85 ^b^ *
N of episodes per day	0.99 ^c^ *	0.88 ^b^ *
Minutes per episode	1.00 ^c^ *	0.93 ^b^ *
Duration per day	0.99 ^c^ *	0.92 ^b^ *
** *Phenomenology Items* **		
Urge	1.00 ^b^ *	0.85 ^b^ *
Gratification	1.00 ^b^ *	0.95 ^b^ *
Awareness	1.00 ^c^ *	0.82 ^c^ *
** *Course of SPD Items* **		
Age at onset	1.00 ^c^ *	0.95 ^c^ *
Abstinence	1.00 ^c^ *	0.91 ^c^ *

* *p* < 0.05; ^a^ = kappa statistic; ^b^ = prevalence-adjusted kappa statistic; ^c^ = Intra-class coefficient.

**Table 4 ijerph-19-06717-t004:** Convergence and Divergence Validity.

	DISP Items/Indices			
	DSM-5 Diagnosis ^1^	Episodes Per Day	Minutes Per Episode	Total Duration
** *Convergence* **				
SPS-R Impairment	0.42 **	−0.10	0.22 *	0.06
SPS-R Duration Item	0.03	0.40 **	0.14	0.47 **
** *Divergence* **				
Stress	0.21 *	0.15	0.01	0.19
Anxiety	0.12	0.16	−0.07	0.19
Depression	0.20 *	0.19	−0.08	0.08
OC symptoms	−0.07	0.02	0.04	0.14
Skin Complaints	0.02	0.18	−0.10	−0.14

^1^ Biserial Correlation Coefficient; * *p* < 0.05; ** *p* < 0.01

**Table 5 ijerph-19-06717-t005:** Comparison of clinical features across different primary picking sites ^1^.

	Primarily Fingers (1) *n* = 45	Primarily Face (2) *n* = 33	Primarily Body (3) *n* = 26	F	Post Hoc (Tukey)
** *% Devoted to Area* **					
Percent Fingers	86.8 (16.9)	4.3 (18.8)	3.1 (9.2)	336.6 ***	1 > 2, 2 = 3, 1 > 3
Percent Face	6.2 (16.3)	79.1 (23.2)	15.0 (18.1)	131.9 ***	1 < 2, 2 > 3, 1 = 3
Percent Body	9.0 (14.3)	15.9 (17.2)	81.5 (18.9)	174.8 ***	1 = 2, 2 < 3, 1 < 3
** *Diagnostic Items* **					
Excessive picking	100 (0.0)	100 (0.0)	100 (0.0)	---	---
Skin damage	89.1 (31.5)	96.7 (18.3)	92.6 (26.7)	0.7 ^ns^	---
Desire to stop	91.9 (27.7)	100 (0.0)	95.0 (22.4)	0.9 ^ns^	---
Attempts to stop	73.9 (44.4)	76.7 (43.1)	92.6 (26.7)	1.9 ^ns^	---
Emotional distress	69.6 (46.5)	90.0 (30.5)	85.2 (36.2)	2.8 ^ns^	---
Impairment	23.9 (43.1)	63.3 (49.0)	55.6 (50.6)	7.6 **	1 < 2, 2 = 3, 1 < 3
DSM-5 Diagnosis	58.7 (49.8)	73.3 (44.9)	81.5 (39.6)	2.3 ^ns^	---
** *Symptom Presentation* **					
N of picking sites	1.8 (0.9)	2.4 (1.1)	2.7 (1.4)	6.3 **	1 < 2, 2 = 3, 1 < 3
N of Episodes	10.5 (11.4)	3.6 (5.5)	5.4 (5.4)	6.3 **	1 < 2, 2 = 3, 1 < 3
Min per episode	9.9 (11.2)	15.6 (17.6)	23.9 (31.1)	4.1 *	1 = 2, 2 = 3, 1 < 3
Total duration	61.8 (61.9)	46.6 (73.5)	78.3 (97.4)	1.1 ^ns^	---
** *Phenomenology* **					
Urge	79.1 (41.1)	72.4 (45.5)	84.6 (36.8)	0.6 ^ns^	---
Gratification	78.6 (41.5)	89.7 (30.9)	96.0 (20.0)	2.3 ^ns^	---
No Awareness	61.7 (26.7)	34.4 (28.9)	39.1 (34.1)	8.7 ***	1 < 2, 2 = 3, 1 < 3
** *Course of the Problem* **					
Age at onset	10.6 (4.4)	12.7 (5.3)	11.1 (6.4)	1.5 ^ns^	---
Abstinence	38.5 (116.5)	33.3 (52.2)	62.1 (162.9)	0.4 ^ns^	---
** *Questionnaires* **					
SPS-R Severity	8.3 (3.1)	6.8 (2.1)	8.1 (2.1)	3.2 *	1 > 2, 2 = 3, 1 = 3
SPS-R Impairment	3.0 (1.9)	4.6 (2.5)	4.7 (2.9)	5.7 **	1 < 2, 2 = 3, 1 < 3
Stress	18.4 (9.9)	15.8 (10.9)	18.7 (9.8)	0.8 ^ns^	---
Anxiety	10.4 (9.4)	8.6 (6.7)	11.8 (9.9)	0.9 ^ns^	---
Depression	15.1 (11.5)	9.9 (9.5)	16.2 (12.7)	2.8 ^ns^	---
OC symptoms	17.9 (16.4)	15.6 (13.2)	17.6 (14.0)	0.3 ^ns^	---
Skin Complaints	7.3 (1.5)	7.6 (1.9)	7.9 (2.6)	0.6 ^ns^	---

^1^ two individuals had incomplete data regarding picking sites; * *p* < 0.05; ** *p* < 0.001; *** *p* < 0.0001; ^ns^ = Not significant.

## Data Availability

The data that support the findings of this study are openly available in Open Science Framework data repository at https://osf.io/t6ycb/.

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
