# Peer review of "Assessing Excoriation (Skin-Picking) Disorder: Clinical Recommendations and Preliminary Examination of a Comprehensive Interview"

_ijerph, 2022, doi:10.3390/ijerph19116717_

Round 1

Reviewer 1 Report

This study developed a new comprehensive interview protocol for diagnosing and classification of excoriation disorder (skin picking disorder). The new tool is a refinement to the broad criteria used in the DSM-V. The study is well carried out, the psychometrics of the tool sound and clearly described. The manuscript is also well-written and is almost ready to be published right away.

I have several suggestions for improvements:

  1. In the Introduction section, three paragraphs were devoted to the possible areas of improvement for better diagnosing excoriation disorder [Lines 53-81]. It would be better to cite more literature to support some of the claims, such as Lines 58-61 and Lines 68-72. In the current form, some claims felt more like personal opinions of the authors or those more appropriate in consulting reports. Rather, to usher in the new tool, it should be identifying shortcomings widely documented or those apparent once examined against certain diagnostic principles or rules.
  2. The relative strength of the new tool could be better illustrated in added predictive power for identifying excoriation symptoms, comorbidities or its general prognostics. Authors may consider collecting more data and conduct regressions to examine the predictive validity.
  3. The suitability of this tool in other populations should be examined in future study, and this is critical because the entire study was implemented using university students, which also requires some justification.
  4. In discussion section, it would be advisable to add briefly the highlights of the new interviewing tool in the first paragraph; i.e.,  how does it differ from DSM-V, etc.
  5. Lines 26-32 should be removed.

Author Response

See word file attached. 

Reviewer 2 Report

ijerph-1729939.  Diagnosing and Assessing Excoriation (Skin-Picking) Disorder: Clinical Recommendations and Development of a Comprehensive Interview

This is a research work of interest. Relevant because there are not many works on Skin-Picking Disorder (SPD) or Excoriation Disorder. It is a clear manuscript, correctly expressed, precise in its delimitation, with adequate bibliographic review, and analysis of the results.

Some considerations will be made that may contribute to detail some of the issues presented in the paper. However, the general idea of the article and its usefulness are not altered by these considerations, and the authors are congratulated for their contribution.

One of the strengths of the paper is that, in addition to providing an instrument for the assessment of SPD, recommendations are made to take into account during this assessment process, such as taking into account not only the overall time spent in the excoriation behavior, but the length and number of episodes, as well as the time spent on each part of the body where this behavior is appreciated.

However, it often happens with studies related to impulsive (and compulsive) behaviors that give the impression of being decontextualized behaviors. We do not know if there is any antecedent of these episodes, from a certain emotional state, some concrete event from which this behavior starts, or if it is a repetition of episodes because the person when left alone tends to focus his attention on one (or several) part(s) of the body (like the one who turns on the television every day), or while the person remembers what happened during the day, or while talking on the phone, or when the person isolates himself because of the daily difficulties in his family...

Nor do we know anything about the exploration in terms of family context background (similar cases, types of repetitive behaviors) or behaviors that serve to regulate mood or improve (apparently) control: OCD, shopping, eating behavior, substances....

Although briefly, some commentary on topographically very different behaviors, such as trichotillomania, but with similar functionality, is missing. In some cases of trichotillomania, we have noticed that the person, in addition to pulling out hair from the head, sometimes pulls out hair from the legs, for example. To do this, he locates areas where the hair is emerging, squeezes the skin, pulls out the incipient hair, and also makes a wound. In some cases, the wound formed serves for the person to initiate checks, scratching, etc. Perhaps some of these combined expressions deserve some consideration as well. At times, we seem more concerned with an exclusionary description, rather than a close or equivalent functional understanding.

It would be interesting, albeit a brief reference, to know not only to what extent these behaviors can have negative consequences, but also the essence of their origin and cause, given that some aspects suggested by the article (and the instrument) seem to point more to the maintenance of the behavior than to its cause and origin. In this way, the usefulness would not only be focused on the evaluation and the scarce knowledge we still have of this type of disorders, but could also guide the intervention. If not from the construction of the instrument, because it has already been designed in a certain way, it can, on the other hand, among possible suggestions for improvement, additions, or for further studies.

The description of the sample lacks some minimum description of the socioeconomic environment.

A minimal recommendation refers to the observation of the reliability of the instruments. Although it is often observed in many publications that such a scale is reliable, and the value of Cronbach's alpha is added, the truth is that this statistic is for the scores, not for the scale.

Among the limitations of the work, it should be pointed out that the sample size, the type of sample chosen, and the selection procedure are limitations. Probably, this work needs a much larger sample size, as well as a sample chosen in a clinical setting, to verify its usefulness with other instruments. Finally, a broader inter-rater comparison should have been performed to more adequately verify the records obtained with the instrument.

The authors are congratulated for this contribution, and are encouraged to consider broader aspects in the phenomenological consideration of SPD.

Author Response

See word file attached. 

Reviewer 3 Report

Thank you for the opportunity to review this study entitled “Diagnosing and Assessing Excoriation (Skin-Picking) Disorder: Clinical Recommendations and Development of a Comprehensive Interview.” (ijerph-1729939).

The study focused on the development of the Diagnostic Interview for Skin Picking Problems (DISP), a new instrument based on the DSM-5 criteria for SPD for the assessment of several clinical features of SPD, including the frequency and duration of picking episodes, and the proportion of time devoted to picking at different body areas.  

In my opinion, the research topic is relevant, and the study is interesting. Parallelly, there are some issues that need to be addressed before the paper will be suitable for publication.

  • Abstract: the information about the sample should be deepened (Mean age and SD? Percentage of men and women?) to provide a clear picture of what will be presented in the paper.
  • Section “0. How to Use This Template” should be deleted.
  • The methodology section requires much more information on how the interview was developed from item creation to initial review and to the piloting reported in the manuscript. It is unclear if the interview has undergone any review or revision before the reported analysis.
  • The “Conclusions” section should be deepened and enriched with the practical implication and highlighting the utility of this new interview.

Author Response

See word file attached. 
